# PPM EMAT for Defect Detection in 90-Degree Pipe Bend

**DOI:** 10.3390/ma15134630

**Published:** 2022-07-01

**Authors:** Linhao Wang, Jiang Xu, Dong Chen

**Affiliations:** School of Mechanical Science and Engineering, Huazhong University of Science and Technology, Wuhan 430074, China; linhaowang@hust.edu.cn (L.W.); chendong_@hust.edu.cn (D.C.)

**Keywords:** PPM EMAT, guided wave, pipe bend, aluminum pipe, defect

## Abstract

Aircraft pipelines are mainly used for the storage and transportation of fuel, hydraulic oil and water, which are mostly bent pipes of non-ferromagnetic materials. We used PPM (Periodic Permanent Magnet) EMAT (Electromagnetic Acoustic Transducer) to detect the defects at 90-degree bends. A simulation model was established by finite element software to study the propagation characteristics and defect detection capability of T (0, 1) mode-guided wave in aluminum pipe bend. In terms of propagation characteristics, the energy of the guided wave was focused in the extrados of the bend, and the guided waves in the intrados and extrados of the bend were separated due to the difference in propagation distance. Regarding defect detection capability, T (0, 1) mode-guided wave had the highest detection sensitivity for the defect in the extrados of the bend and the lowest detection sensitivity for the defect in the middle area of the bend. We designed a PPM EMAT for 320 kHz to verify the simulation results experimentally, and the experimental results are in good agreement with the simulation results.

## 1. Introduction

Aircraft pipelines, as shown in Figure 1, are mainly used to store and transport fuel, lubricating oil, hydraulic oil, water and air. It mainly consists of bent pipes of non-ferromagnetic materials such as titanium alloy, stainless steel and aluminum alloy. Due to vibration, corrosion, fatigue damage or potential internal defects of the pipelines, the pipeline system on the in-service aircraft may cause pipeline leakage or rupture, affecting the reliability and safety of the aircraft system. There are many 90-degree bends in the pipeline system. Due to stress concentration, bends are easily defective. It is necessary to study detecting defects in 90-degree pipe bends.

Sanderson et al. conducted a lot of theoretical research on the propagation of guided waves in bends [1,2,3,4]. Demma used the modal analysis method to analyze the dispersion curve of guided waves in bends [5,6]. Nishino studied mode conversions from T (0, 1) to other higher-order torsional modes in welded bends [7]. Furukawa used finite elements to study the mode conversion of T (0, 1) mode-guided wave at the bend [8]. Nishino used laser ultrasound to study the propagation of L (0, 1) mode-guided waves at bends [9]. Verma analyzed the Mode conversion of the L (0, 2) mode-guided wave after passing through the bend [10]. Predoi learned the scattering characteristics of the L (0, 2) mode-guided wave at the bend [11]. In recent years, much research has been conducted on defect detection in pipe bends. Yamamoto [12] used piezoelectric sensors to excite low-frequency guided waves below 50 kHz to detect defects in the bend of welded aluminum pipes. Qi [13] used piezoelectric sensors to excite 75 kHz low-frequency guided waves to detect defects in carbon steel bends. He [14] used piezoelectric sensors to excite L (0, 2) guided waves with a frequency of 50 kHz to detect iron bends. Chen [15] and Zhu [16] used magnetostrictive guided waves at about 40 kHz to detect defects in the bend of carbon steel pipes. Simonetti [17] used 32 channels of EMAT to pipe bends with welded seams at a frequency of 130 kHz. Liu and Ding [18,19] used magnetostrictive strip sensors to detect defects on stainless steel bends, both of which used detection frequencies below 100 kHz. Most aircraft pipes are non-ferromagnetic materials. When magnetostrictive sensors are used, magnetostrictive strips are required to excite guided waves. Piezoelectric sensors require coupling agents in use, which increases the practical difficulty of detection. The longer wavelength results in lower detection accuracy of low frequency guided waves compared to high frequency guided waves. EMAT (Electromagnetic Acoustic Transducer) has the advantages of easy excitation, no coupling agent, flexible detection method, and high-temperature resistance. It usually has a high detection frequency, resulting in high detection accuracy. In summary, EMAT can be used to detect the defects in non-ferromagnetic pipe bends [20,21,22,23,24,25,26,27,28,29,30]. T (0, 1) mode guided wave is one of the most widely used pipe detection guided waves due to its advantages of no dispersion and low attenuation in pipes transporting liquids. PPM (Periodic Permanent Magnet) EMAT (Electromagnetic Acoustic Transducer) is currently used mainly to excite T (0, 1) mode-guided waves. With this transducer, T (0, 1) mode-guided waves can be excited in non-ferromagnetic pipes [31,32,33,34,35,36,37,38].

PPM EMAT is mainly suitable for detecting stainless steel, carbon steel and aluminum pipes in the industry. In this paper, PPM EMAT was used to detect defects in the bend of aluminum pipes by simulations and experiments. The results show that when T (0, 1) mode guided wave propagates through the bend, the intrados and extrados waves are separated after the guided wave propagates for a certain distance because the propagation paths are different. Most of the energy of the guided wave is concentrated in the extrados of the bend. T (0, 1) mode-guided wave has the highest detection sensitivity for the defect in the extrados of the bend and the lowest detection sensitivity for the defect in the middle area of the bend.

## 2. Finite Element Simulations

The propagation of the guided wave in the bend was simulated by COMSOL5.5 software. The material used in the simulation model was 6063-grade aluminum alloy with a density of 2690 kg/m^3^, Poisson’s ratio of 0.33 and Young’s modulus of 69 Gpa. In order to control the errors of waveform propagation to within 5%, the maximum grid size was set to one-eighth the wavelength [13]. The grid at the defect was refined to one-sixtieth of a wavelength. A three-period sinusoidal signal modulated with a Hanning window was generated at one end of the pipe with a frequency of 320 kHz in the circumferential direction. The outer diameter of the pipe was 30 mm, and the wall thickness was 1.5 mm. The layout of the simulation is shown in Figure 2a. Based on the propagation time of T (0, 1) mode-guided wave in the pipe, the computation time was set to 0.2 ms to receive the passing signal. The meshed calculation contained 2,961,543 degrees of freedom. The grid division diagram is shown in Figure 2b.

### 2.1. Propagation Characteristics

In order to study the propagation characteristics of the T (0, 1) mode-guided wave through the pipe bend, the signal was excited at one end of the model and received the passing signal at the other end. The received passing signal is shown in Figure 3. From the figure, we can see that when T (0, 1) mode-guided wave propagates through the pipe bend, the separation of the intrados wave and the extrados wave occurs. This is because the propagation paths are different. The intrados wave expands to the extrados, forming a new wave.

The energy distribution diagram is shown in Figure 4. It can be seen from the cloud diagram that most of the energy of the guided wave was concentrated in the extrados of the bend. The phenomenon of intrados and extrados wave separation and uneven energy distribution of guided waves has an impact on defect detection at different circumferential positions of the bend. Therefore, we needed to perform finite element simulations of defective bends to verify the ability of T (0, 1) guided waves to detect defects in the bend.

### 2.2. Defect Detection Capability

In order to verify the ability of the T (0, 1) guided wave to detect defects at the bend, a notch defect was simulated by removing elements. A defect of 10 mm long, 0.5 mm wide and 1 mm deep was created, as shown in Figure 5a,b. We arranged the defect at 45° of the pipe bend. Defects in the intrados, middle area and extrados of the bend were removed, respectively. The receiving position was set to receive the defect echo signal, and its layout is shown in Figure 5c.

From the simulation results in Figure 6, it can be concluded that the echo amplitude of the defect located in the extrados of the bend is the largest, and the echo amplitude of the defect located in the middle area of the bend is the smallest. This shows that T (0, 1) guided wave can detect the defects at each position of the bend circumference, but the sensitivity to the defects at different positions is different.

## 3. Principles and Geometric Parameters of the PPM EMAT

### 3.1. Principles of PPM EMAT

Standard PPM EMAT consists of several racetrack coils and arrays of magnets with periodic polarities, as shown in Figure 7. The eddy currents in the specimen induced by the AC coil will experience Lorentz forces in both the static magnetic field from the magnets and the dynamic magnetic field from the AC coil. The alternating Lorentz force will generate periodic vibrations that form a horizontally polarized shear wave (SH). The AC current of the sensor is passed through the racetrack coil, and the Lorentz force (*F_L_*) is the body force per unit volume generated by the interaction between the induced eddy current (*J_e_*) and the magnetic field (*B_s_*). This relationship is shown in Equation (1). Compared to the static magnetic field (*B_s_*) generated by the permanent magnet, the dynamic magnetic field generated by the AC current is negligible [39,40].
*F_L_* = *J_e_* × *B_s_*(1)

For PPM EMAT arrays in metal pipes, we needed a radial magnetic field and an axial excitation current to excite T (0, 1) mode-guided waves. The axial current and radial magnetic field generate a circumferential Lorentz force, which excites T (0, 1) mode-guided waves propagating in the axial direction of the pipe. Figure 8 shows the mechanism of using an array of PPM EMAT to generate a circumferential Lorentz force in a metal pipe to excite T (0, 1) guided waves.

Several PPM EMAT elements form a circular array around the pipe. Each element consists of a racetrack coil and two PPM arrays. Two straight sections of the racetrack coil were placed along the axis of the pipe, on which two PPM arrays were placed. A racetrack coil carries two current parts in opposite directions. Axial eddy currents generate circumferential Lorentz forces in radial static magnetic fields. By controlling the polarization direction of the magnet and the excitation current direction of different elements, the eddy currents on the cross-section are subjected to the Lorentz force in the same circumferential direction. Circumferential alternating vibrations will propagate along the pipe, thus producing T (0, 1) mode-guided waves [31].

### 3.2. Geometric Parameters of the PPM EMAT

The geometric parameters of PPM EMAT are shown in Figure 9. According to the references [13], the geometric parameters of the PPM EMAT coil mainly include the length of the coil, *l_c_*; the width of the coil, *w_c_*; the distance between adjacent racetracks, *w_g_*_1_; the distance between the outermost racetrack and the coil, *w_g_*_2_; the distance between adjacent coils, *w_g_*_3_; thickness, *h_r_*; and width, *w_r_*, of the racetrack. The parameters of the coil are shown in Table 1.

The parameters of the magnets include not only the width of the magnets, the height of the magnets, the length of the magnets and the number of magnet arrays but the placement relationship between the magnets. We used N52 grade magnets and arranged them in an array of six for placement on the six racetracks. The current direction of adjacent coils and the position of the magnets affect the amplitude of the signal. We adopted the method with the largest signal amplitude. This arrangement is to have the current direction of the adjacent coils be opposite, and the direction of the magnet array be exactly opposite. Since the coils used in the experiments consisted of three racetrack coils, we set the current direction of the middle coil to be opposite to that of the coils on both sides and placed the magnet array in the opposite direction [31]. The periodic permanent magnet array and racetrack coil are shown in Figure 10.

## 4. Experimental Verification

### 4.1. Experimental Setup

In order to verify the performance of the transducer, we conducted experiments on a 6063-grade aluminum pipe with an outer diameter of 30 mm, a wall thickness of 1.5 mm and a length of 1197.8 mm. By calculation, Figure 11 shows the group speed dispersion curve of this pipe. The excitation frequency used during the experiment is 320 kHz, so the guided wave group velocity is about 3066 m/s according to the dispersion curve. There are artificial notch defects and wear defects on the pipe, as shown in Figure 12. The notch defect depth in intrados of the bend is 0.8 mm, and the loss of the defect cross-sectional area is about 3.857%; the defect depth in the middle area of the bend is 0.8 mm, and the defect cross-sectional area loss is about is 3.857%; the defect depth in extrados of the bend is 0.2 mm, and the loss of the defect cross-sectional area is about 0.484%. The wear defect depth in intrados of the bend is 0.6 mm, and the loss of the defect cross-sectional area is about 2.512%; the defect depth in the middle area of the bend is 0.6 mm, and the defect cross-sectional area loss is about is 2.512%; the defect depth in extrados of the bend is 0.2 mm, and the loss of the defect cross-sectional area is about 0.484%.

We used the laboratory’s own electromagnetic ultrasonic instrument for experiments. The signal of this instrument was amplified by about 10,000 times. The pass frequency of the bandpass filter was 200–400 kHz, and the sampling rate was 50 MHz. The excitation signal was a three-cycle sine wave current. In order to reduce noise, each signal was sampled 100 times to obtain an average value. In order to verify the simulation results of T (0, 1) guided wave passing through the bend, we named the experiment Experiment 1 to receive the passing signal on a defect-free bend. We designated the experiment performed without defects as Experiment 2; the experiment performed with a defect in the intrados of the bend as Experiment 3; the experiment performed with a defect in the middle area of the bend as Experiment 4; and the experiment performed with a defect in extrados of the bend as experiment 5. We named Experiments 6, 7 and 8, which were performed with wear defects at the three locations. The experimental layouts are shown in Figure 13.

### 4.2. Experimental Data

From Figure 14, we can see that the received passing signal is composed of two waves. The first one is a low-energy wave through the intrados of the bend. The second one is a high-energy wave through the extrados of the bend.


(1)Based on the propagation distance, the time for the first wave to reach the receiver should be 0.2819 ms, and the time for the second wave to reach the receiver should be 0.2972 ms. From the received signal, the time for the first wave to reach the receiver is 0.2833 ms and the time for the second wave to reach the receiver is 0.3011 ms;(2)The peak-to-peak value of the first wave is 1.2038 V and the peak-to-peak value of the second wave is 7.399 V. The amplitude ratio of the first wave to the second wave is 16.33%. The amplitude ratio of the first wave to the second wave in the simulated signal is 14.35%.


The signals received in the notch defect detection experiments are shown in Figure 15. According to the calculated dispersion curve, when the excitation frequency is 320 kHz, the group velocity of the T (0, 1) mode-guided wave is about 3066 m/s. According to the calculation, the end echo should appear at a time of 0.2674 ms. In the waveforms of the four signals, the echoes appear at 0.2740 ms, 0.2752 ms, 0.2734 ms and 0.2726 ms, respectively. The calculated wave speeds are about 2993 m/s, 2980 m/s, 3000 m/s and 3008 m/s, respectively. It can be concluded that the waves excited by the transducer are all T (0, 1) mode-guided waves. The amplitude of the defect echo is as follows:(1)When the defect is located in the intrados of the bend, the peak-to-peak value of the defect echo is about 0.5683 V;(2)When the defect is located in the middle area of the bend, the peak-to-peak value of the defect echo is about 0.2352 V;(3)When the defect is located in the extrados of the bend, the peak-to-peak value of the defect echo is about 0.6731 V.

In the four signals, the peak-to-peak values of the passing signals are 12.049 V, 11.926 V, 11.953 V and 11.946 V. The ratio of the peak-to-peak value of defect echo to the peak-to-peak value of the first passing wave are about 4.76%, 1.96% and 5.63%, respectively.

The signals received in the wear defect detection experiments are shown in Figure 16. In the waveforms of the four signals, the echoes appear at 0.2740 ms, 0.2755 ms, 0.2758 ms and 0.2730 ms, respectively. The calculated wave speeds are about 2993 m/s, 2976 m/s, 2973 m/s and 3003 m/s, respectively. It can be concluded that the waves excited by the transducer are all T (0, 1) mode-guided waves. The amplitude of the defect echo is as follows:(1)When the defect is located in the intrados of the bend, the peak-to-peak value of the defect echo is about 0.3467 V;(2)When the defect is located in the middle area of the bend, the peak-to-peak value of the defect echo is about 0.2528 V;(3)When the defect is located in the extrados of the bend, the peak-to-peak value of the defect echo is about 0.4581 V.

In the four signals, the peak-to-peak values of the passing signals are 12.049 V, 11.982 V, 12.043 V and 12.085 V. The ratio of the peak-to-peak value of defect echo to the peak-to-peak value of the first passing wave are about 2.89%, 2.10% and 3.79%, respectively.

### 4.3. Results and Discussions

In Figure 14, there are two waves with different energy. The first wave has small energy, while the second wave has large energy. The time of their arrival at the receiver is very close to the theoretical calculation. The amplitude ratio of the first wave to the second wave is 16.33%, which is very close to the 14.35% amplitude ratio of the first wave to the second wave in the stimulation signal. Therefore, T (0, 1) mode-guided wave is separated from intrados and extrados waves after passing through the bend. The energy of the wave is concentrated in the extrados of the bend. Due to the difference in the propagation path, the intrados wave is received before the extrados wave.

In Figure 15 and Figure 16, by comparing the defect echoes in intrados, middle area and extrados of the bend, it can be concluded that our transducer is the most sensitive to the defect in extrados of the bend and the worst to the defect in the middle area of the bend, which is consistent with our simulation results. The cross-sectional area losses of the three notch defects are 3.857%, 3.857% and 0.484%, respectively. The ratios of the peak-to-peak value of notch defect echo to the peak-to-peak value of passing signal are 4.76%, 1.96% and 5.63%, respectively; The cross-sectional area losses of the three wear defects are 2.512%, 2.512% and 0.484%, respectively. The ratios of the peak-to-peak value of wear defect echo to the peak-to-peak value of passing signal are 2.89%, 2.10% and 3.79%, respectively. The cross-sectional area loss and the ratio of the peak-to-peak value of defect echo to the peak-to-peak value of passing signal are inconsistent, which is mainly because of the uneven energy distribution of T (0, 1) mode guided wave passing through the bend. This phenomenon results in different sensitivities of T (0, 1) mode-guided waves to defects at different circumferential positions of the bend.

From the experiment and simulation results, the PPM EMAT has good sensitivity to the defects in the intrados and extrados of the bend, but the sensitivity to the defects in the middle area of the bend is low. When the noise is large, it is easy to misjudge the defect in the middle area of the bend. When T (0, 1) mode-guided wave propagates through the bend, the energy is focused on the extrados of the bend. The energy in the middle area is greater than the energy in intrados. The echo from the defect in the middle area of the bend should theoretically be larger than that from the defect in intrados. However, the simulation and experiment results are just the opposite. This is mainly because the signal received by the receiver is the average value of the entire circular vibration. Mode conversion occurs when T (0, 1) mode-guided wave propagates through the defect in the middle area of the bend. The defect echo is not an axisymmetric mode wave on the circumference, resulting in a reduction in the defect echo.

## 5. Conclusions and Future Works

In this paper, PPM EMAT was used to detect the aluminum pipe bends through simulations and experiments. From the result, we can conclude:(1)When T (0, 1) mode-guided wave propagates through the bend, the waves in intrados and extrados of the bend are separated due to the different propagation paths;(2)The wave energy is focused on the extrados of the bend, which leads to the different sensitivity of the PPM EMAT to detect defects at different circumferential positions of the bend;(3)The PPM EMAT is most sensitive to defects in the extrados of the bend and least sensitive to defects in the middle area of the bend.

PPM EMAT has almost the same detection capability for artificial notch defects and wear defects. Since the loss of the cross-sectional area of the wear defect is smaller, the defect echo of the wear defect is smaller than that of the artificial notch defect. However, for defects at different locations, PPM EMAT has the same sensitivity for both types of defects. When T (0, 1) guided wave propagates through a defect in the middle area of the bend, a mode conversion occurs, and T (0, 1) mode guided wave changes to an asymmetric mode. The signal received by the receiver is the average of the whole circular vibration, which results in the defects in the middle area of the bend not being easily detected.

In future applications, mode conversion occurs when the T (0, 1) mode-guided wave propagates through a defect in the middle area of the bend will be studied. A focused PPM EMAT will be used to improve the detection capability of the transducer. The positioning of the defect in the circumference position of the bend will also be studied to determine the exact position of the defect.

## Figures and Tables

**Figure 1 materials-15-04630-f001:**
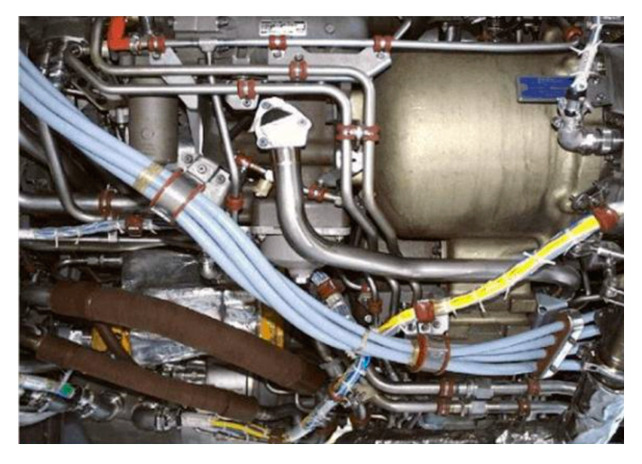
Photo of the aircraft pipeline system.

**Figure 2 materials-15-04630-f002:**
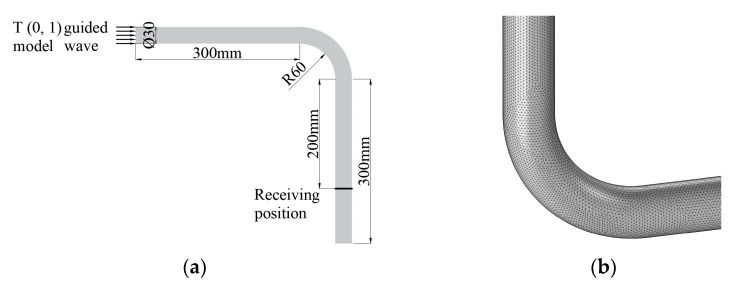
Layout and grid diagrams for simulations. (**a**) Simulation layout for passing signal; (**b**) Meshing diagram.

**Figure 3 materials-15-04630-f003:**
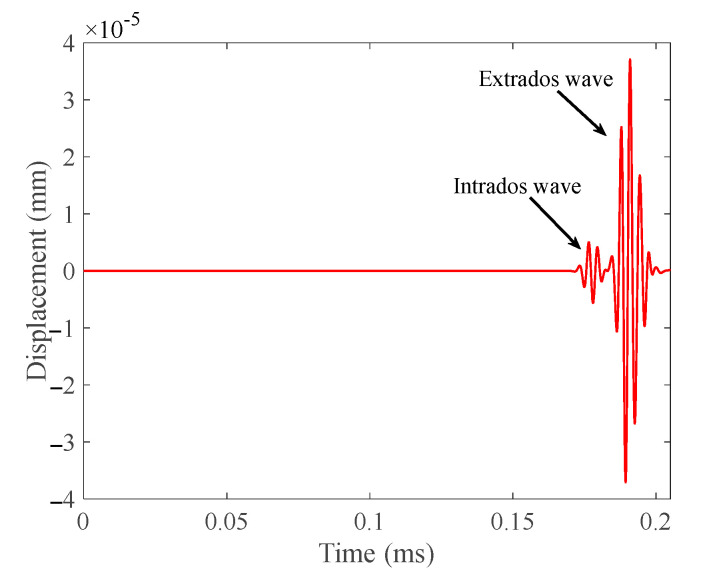
Passing signal received according to the layout in Figure 2a.

**Figure 4 materials-15-04630-f004:**
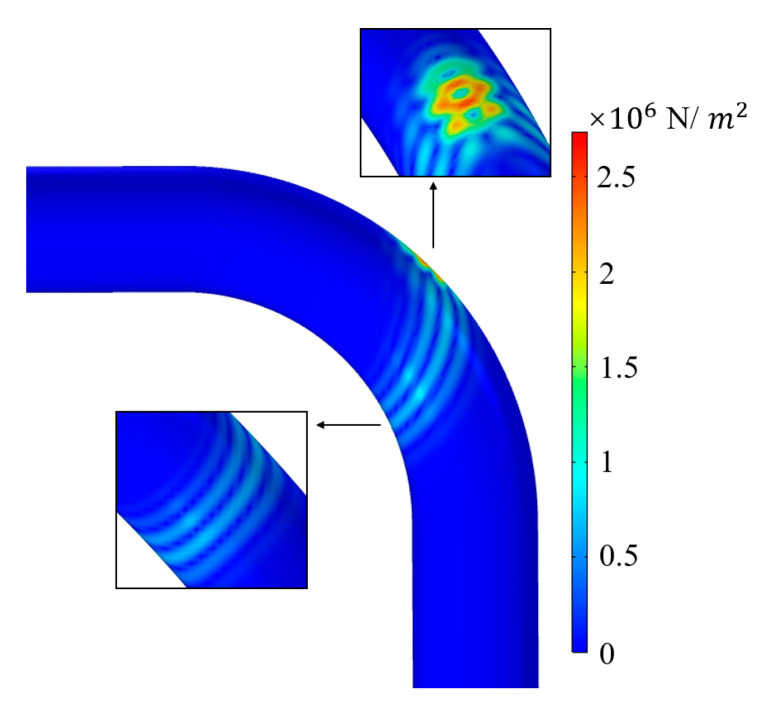
Energy distribution diagram when T (0, 1) guided wave propagates through the bend.

**Figure 5 materials-15-04630-f005:**
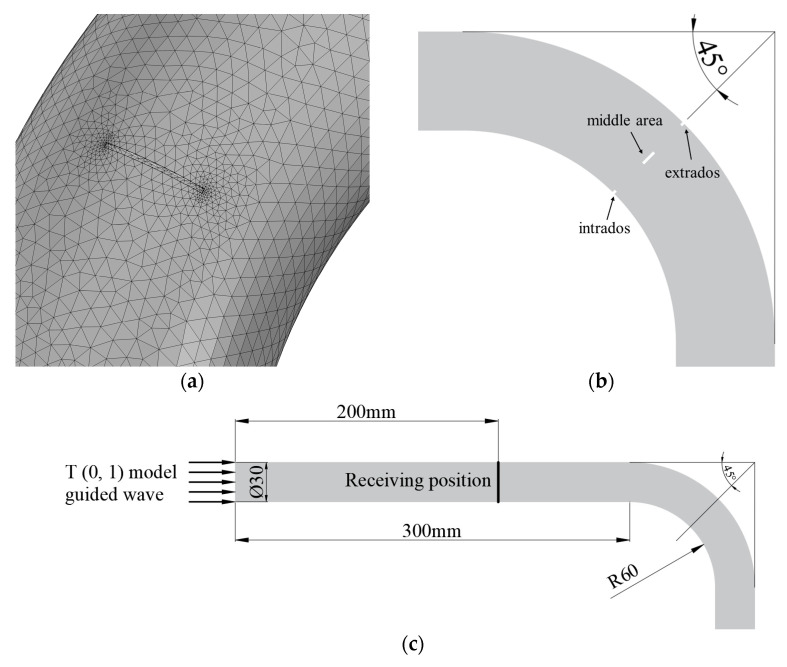
Schematic diagram of finite element mesh of the defect, three positions of the defect in the bend and simulation layout. (**a**) Defect model and meshes; (**b**) Three positions of the defect; (**c**) Simulation layout for defect detection.

**Figure 6 materials-15-04630-f006:**
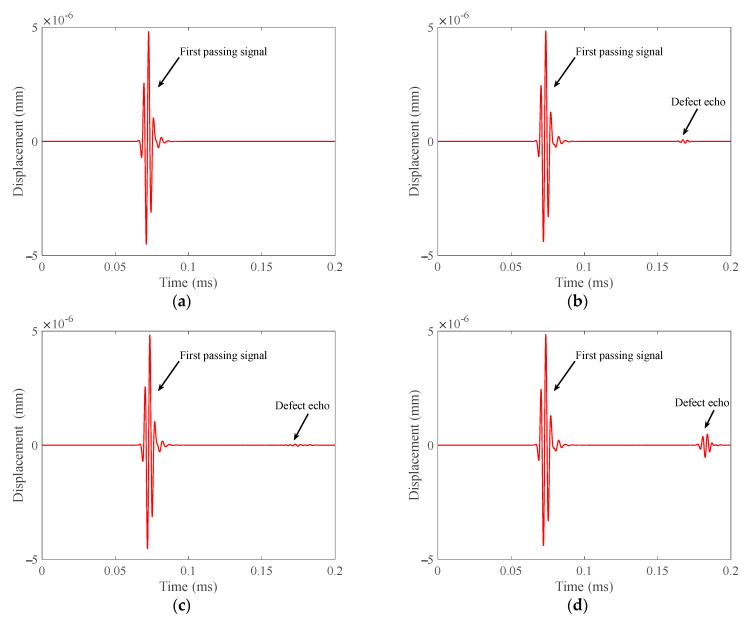
Defect echoes received according to the layout in Figure 5c. (**a**) Defect-free pipe; (**b**) Defect in intrados of the bend; (**c**) Defect in middle area of the bend; (**d**) Defect in extrados of the bend.

**Figure 7 materials-15-04630-f007:**
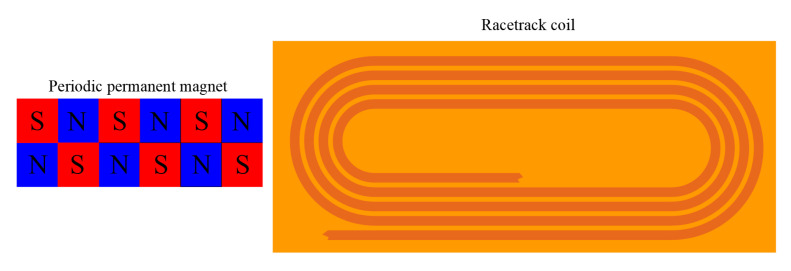
The magnet array and racetrack coil.

**Figure 8 materials-15-04630-f008:**
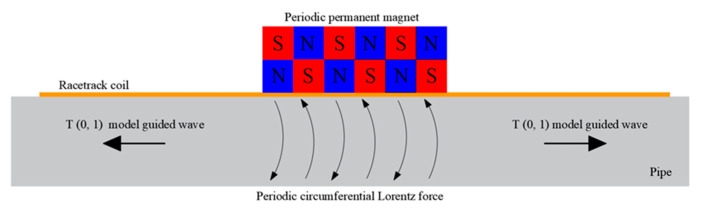
Lorentz mechanism of torsional wave.

**Figure 9 materials-15-04630-f009:**
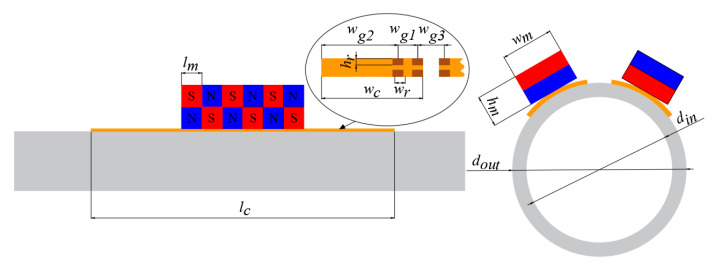
Schematic of the PPM EMAT.

**Figure 10 materials-15-04630-f010:**
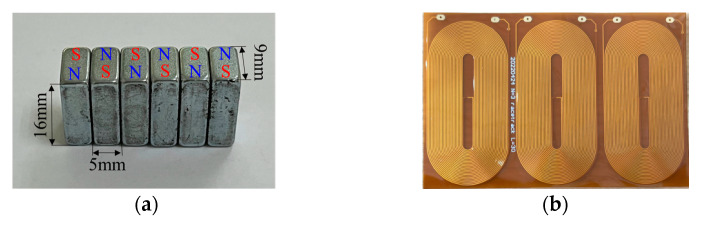
Photos of magnet array and racetrack coil. (**a**) Periodic permanent magnet array; (**b**) Racetrack coil.

**Figure 11 materials-15-04630-f011:**
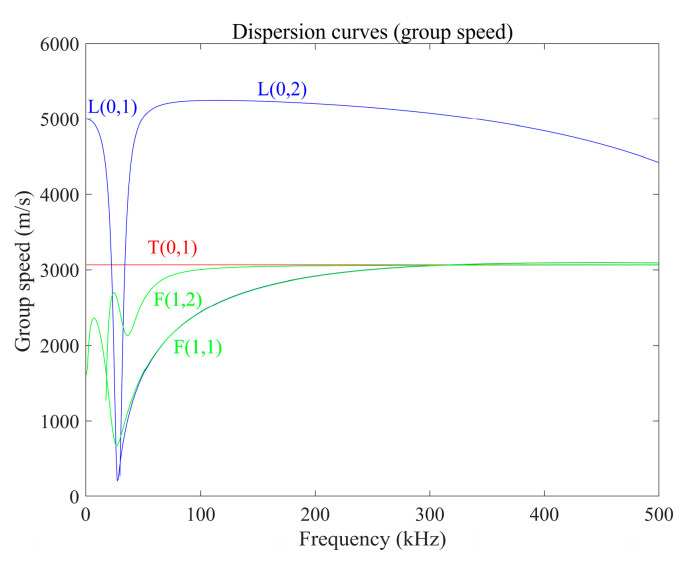
Group speed dispersion curve of an aluminum pipe with an outer diameter of 30 mm and an inner diameter of 27 mm.

**Figure 12 materials-15-04630-f012:**
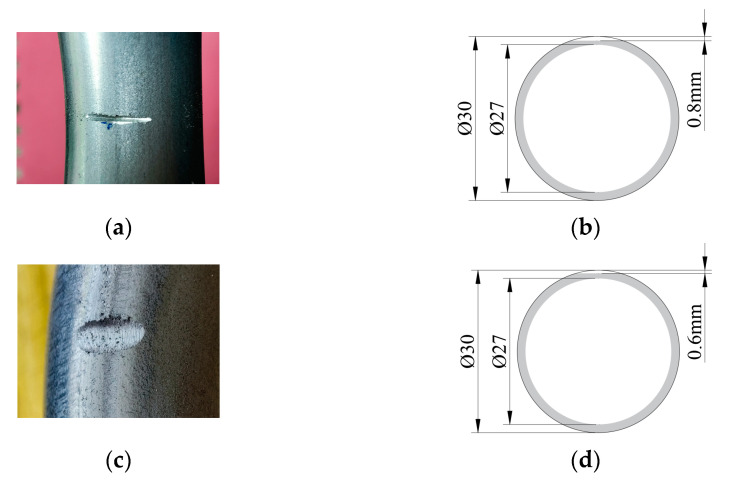
The photo and cross-sectional schematic of artificial defects. (**a**) Notch defect in intrados of the bend; (**b**) Cross-section of the notch defect; (**c**) Wear defect in middle area of the bend; (**d**) Cross-section of the wear defect.

**Figure 13 materials-15-04630-f013:**
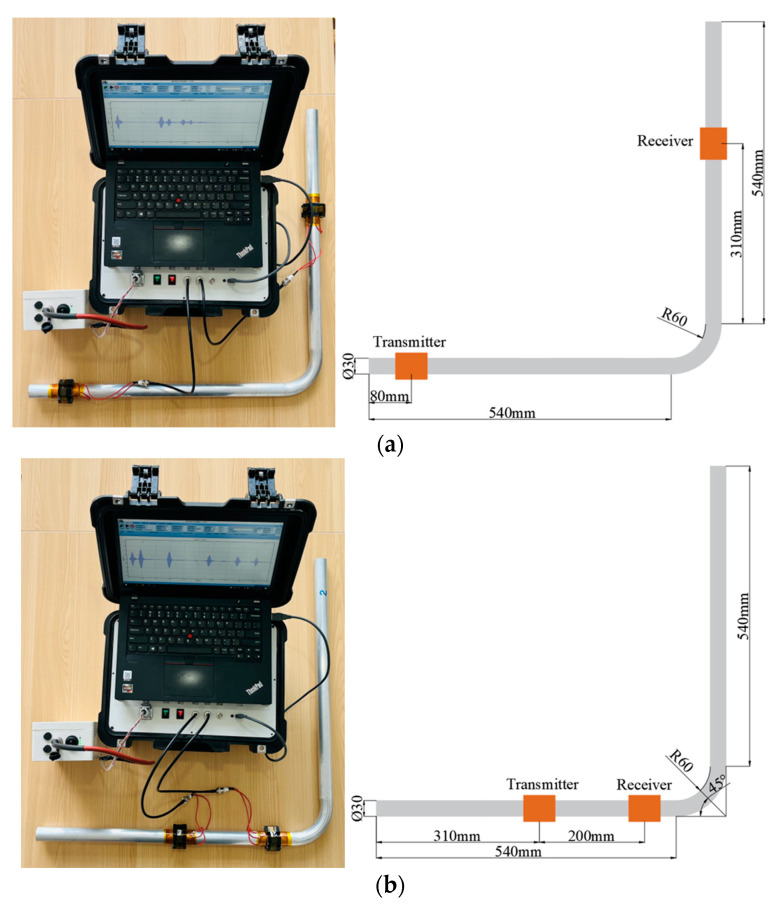
The photos and schematic diagrams of propagation characteristics experiment and defect detection experiment. (**a**) Propagation characteristic experiment; (**b**) Defect detection experiment.

**Figure 14 materials-15-04630-f014:**
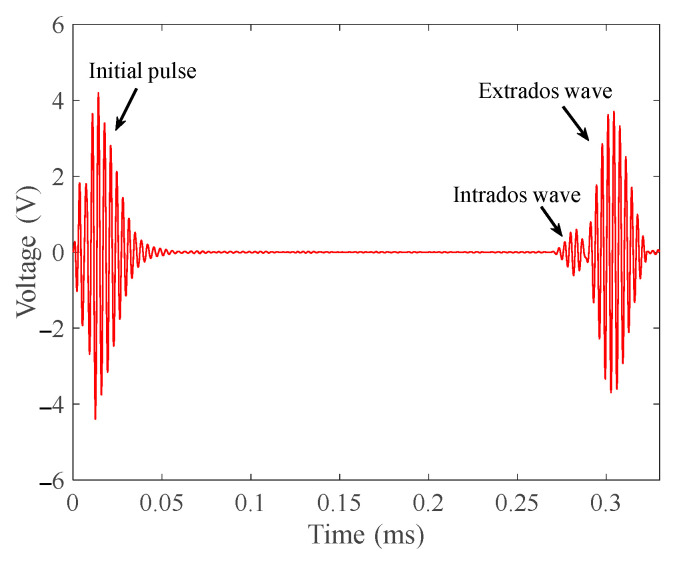
Passing signal received according to the layout in Figure 13a.

**Figure 15 materials-15-04630-f015:**
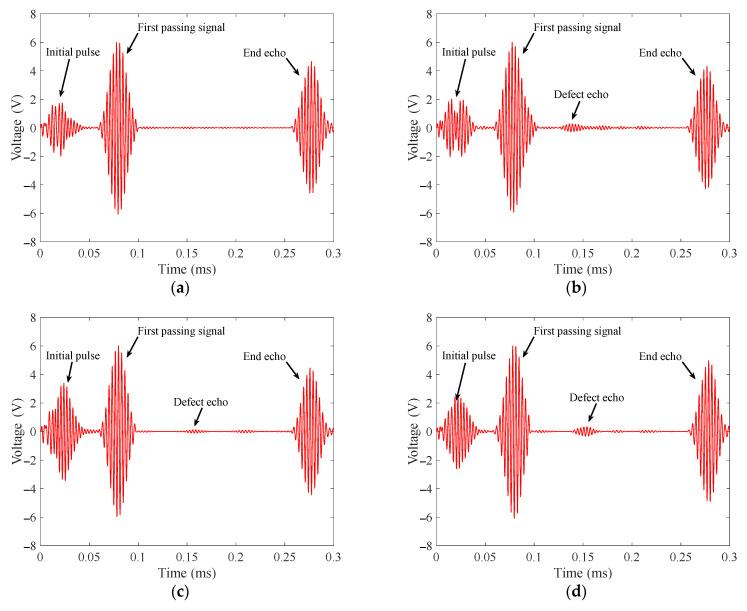
Notch defect echoes received according to the layout in Figure 13b. (**a**) Defect-free bend; (**b**) Notch defect in intrados of the bend with a cross-sectional area loss of 3.857%; (**c**) Notch defect in middle area of the bend with a cross-sectional area loss of 3.857%; (**d**) Notch defect in extrados of the bend with a cross-sectional area loss of 0.484%.

**Figure 16 materials-15-04630-f016:**
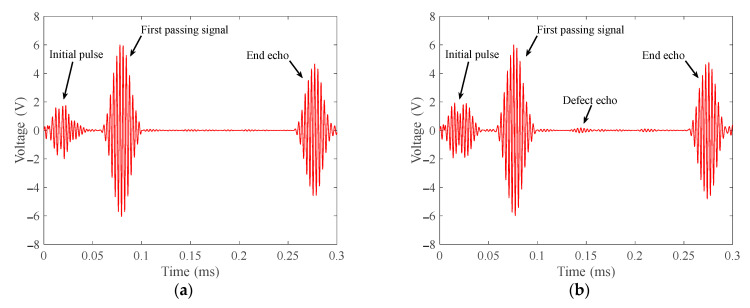
Wear defect echoes received according to the layout in Figure 13b. (**a**) Defect-free bend; (**b**) Wear defect in intrados of the bend with a cross-sectional area loss of 2.512%; (**c**) Wear defect in middle area of the bend with a cross-sectional area loss of 2.512%; (**d**) Wear defect in extrados of the bend with a cross-sectional area loss of 0.484%.

**Table 1 materials-15-04630-t001:** Geometric parameters of PPM EMAT for 320 kHz.

Object	Symbol	Value
Magnet	*l_m_*	5 mm
	*h_m_*	9 mm
	*w_m_*	16 mm
Coil	*l_c_*	60 mm
	*w_c_*	12.19 mm
Racetrack	*w_r_*	10 mil
	*h_r_*	1 oz
	*w_g_* _1_	20 mil
	*w_g_* _2_	67 mil
	*w_g_* _3_	173 mil
Pipe	*d_in_*	30 mm
	*d_out_*	27 mm

## Data Availability

The data supporting reported results by the authors can be sent by e-mail.

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
