# Peer review of "PPM EMAT for Defect Detection in 90-Degree Pipe Bend"

_materials, 2022, doi:10.3390/ma15134630_

Round 1

Reviewer 1 Report

The paper deals with detection of wall defects in a 90-degree aluminium pipe bend. The authors have used PPM (Periodic Permanent Magnet) EMAT (Electromagnetic Acoustic Transducer) method to experimentally excite and detect torsional guided waves (T (0,1) mode) in the pipe with bend. A simulation model based on the finite element method has been used in order to investigate the propagation characteristics of T (0,1) guided wave mode. The authors have designed their own PPM EMAT to validate the simulation results experimentally. The calculated and measured results agree reasonably well. The authors are to be congratulated for their excellent contribution. I recommend the paper to be published in Materials – very nice work indeed.

These are few minor items that need to be considered (addressed) in the final manuscript:

1. Please correct typing errors throughout the paper and enlarge the font size of dimensions in Figs. 2, 4, 9 and 13 (poor visibility).

2. Page 2, Line 66: Please state few typical pipe materials in industry suitable for PPM EMAT NDT.

3. Page 3, Fig. 3 and where applicable: For clarity please provide units for the displacement.

4. Page 3, Fig. 4: For clarity please provide units for energy cloud scale.

5. Page 4, Section 2.2: Please provide the data on mesh density in terms of a number of elements per wavelength, in particular at the three positions of defect.

Reviewer 2 Report

In the Reviewer opinion the research paper entitled “PPM EMAT for defect detection in 90-degree pipe bend” is average.

A simulation model is established by finite element software to study the propagation characteristics of T (0, 1) mode guided wave through aluminum pipe bend. Defects are arranged in intrados, middle area, and extrados of the bend to simulate the detection ability of T (0, 1) guided wave. The results show that T (0, 1) guided wave can detect the defects in 90-degree bend. T (0, 1) guided wave has the highest detection sensitivity for the defect in extrados of the bend and the lowest detection sensitivity for the defect in middle area of the bend. The experimental results are in good agreement with the simulation results very well.

Some comments which greatly enhance the understanding of the paper and its value are presented below. Specific issues that require further consideration are:

1.     The title of the manuscript is matched to its content.

2.              The structure of the manuscript is rather proper.

3.              The Introduction  covers the cases.

4.              In the Reviewer’s opinion, the current state of knowledge relating to the manuscript topic has not been covered and clearly presented.

5.              An analysis of the manuscript content and the References shows that the manuscript under review constitutes a summary of the Author(s) achievements in the field.

6.              Please corrected  all drawings – add name and units of each axis.

7.              Article has flaws, additional experiments needed, research not conducted correctly.

8.              In the Reviewer’s opinion, the bibliography, comprising 30 references, is not representative and exhaustive.

9.              I suggest expanding the conclusions.

10.           In the Reviewer’s opinion the manuscript can be published in the journal, but after major revision.

Reviewer 3 Report

The article is clearly written. The author describe in detail and with interesting scientific details the experiment performed. The introduction is sufficient and the research aim is clear, the methodology is suitable, the results are presented in a proper manner, their discussion is more or less sufficient, the conclusions, in part, represent the main achievements. The article is well structured and the figures are made suggestively and contain important data for the study presented. I honestly have no corrections to make to improve this material. I admit that this field has been and is intensely studied, it does not present a novelty in research, but the way of presenting and the dedication with which this material seems to have been worked deserves my acceptance for publication.  

I recommend the author to check one more time the bibliography. 

Best regards, 

Reviewer 4 Report

The article “PPM EMAT for defect detection in 90-degree pipe bend” is focused on the diagnosis and characterization of bent pipes using 8 Periodic Permanent Magnets and Electromagnetic Acoustic Transducer to detect the defects at 90-degree bends, while FEM model is used to study the propagation characteristics through aluminum pipe bend.  The manuscript is not well designed nor organized. The size and resolution of the figures, although they are wisely selected, do not help the reader follow the scientific description in the text. Extensive editing of English language and style required, since many statements are repeated, and the text becomes confusing due to the grammar and syntax errors. The research work is very interesting and deserves to be published but only after a major revision.

  1. The abstract of the paper is big and info that are not needed are discussed. Therefore, the main purpose and innovation of this work is not highlighted. A revision to focus on the new findings is suggested.
  2. Extensive editing of English language and style is required, and formatting and correction of grammar and syntax errors.
  3. The figures are very well selected but the legends, captions, size and resolution, their explanation and indications using symbols is suggested to highlight the points of discussion clearly.
  4. The presentation of the results must be revised to highlight the scientific findings of the Authors clearly.
  5. Introduction should be better focused in order for the reader to understand why this work is performed. 

Round 2

Reviewer 2 Report

Authors corrected article follow to my suggestion. In my opinion can be published in the Journal.